# Monasticism and Ecologism: Between Economic Opportunity and Religious Convictions?

**Isabelle Jonveaux**

Privat Pädagogische Hochschule, 8010 Graz, Austria; isabellejonveaux@yahoo.fr

**Abstract:** Monasteries, especially Benedictine monasteries, have developed a close relationship with nature based on a respect for creation and a goal of self-sufficiency. There seems to be an elective affinity between monasticism and ecologism. Since the second half of the 20th century, monasteries have been engaging in ecological practices, and in many cases have been pioneers in these practices in their region. How can the role of monasteries in ecology be explained? To what extent is the ecology practiced by monasteries different from that of society? And what developments can we observe in this field over the last 20 years? After presenting the background of the elective affinity between monasticism and ecologism, I will explore the practices that monastics put in place to act sustainably for the protection of the environment. The last part of the paper deals with the shift from respect for creation to ecologism in the vocabulary monastics use today and to what extent we can speak of a charismatic ecologism. This article is based on field inquiries conducted in monastic communities in six countries in Europe and four countries in Africa between 2004 and 2019.

**Keywords:** monasticism; ecologism; Catholicism; sustainable development; Benedictine

## 1. Introduction

Monasteries, especially the Cistercians, are known in European history for having cleared land to make it arable. In France, "it has been calculated that a third of its territory was cultivated by monks" (Martin 1880, p. 82). Male monasteries in the Middle Ages also contained agricultural schools and developed a lot of innovative technics. At first glance, it appears that monasteries have a special relationship with their natural environment. Danièle Hervieu-Léger identifies an elective affinity between ecologism and religious eschatology (Hervieu-Léger 1982). "The concept of Ecologism starts from the position that the non-human world is worthy of moral consideration and that this should be taken into account in the ordering of social, economic and political systems" (Baxter 1999). Ecologism is understood here as an ideology that aims at a better balance between the human being and the natural environment, and the protection of the latter. Ecology is here considered as a synonym of ecologism. Does that mean that there is also an elective affinity between ecologism or sustainable development and monasticism? If so, where does it come from? While monasticism has an age-old relationship with nature, ecology and sustainable development have become highly topical issues in European society. To what extent does the urgency of environmental issues in society influence monasteries?

I am speaking here about monastic orders, not about apostolic orders such as the Franciscans, who have another kind of relationship with nature, which is integrated in their spirituality (Canticle of Brother Sun from Francis of Assisi, for instance). Nevertheless, we will see that not all monastic orders have the same relationship with nature.

After presenting the background of the elective affinity between monasticism and ecologism, I will explore the practices that monastics put in place to act sustainably for the protection of the environment and the role of pioneers that they play in this field. Finally, I will show the shift from respect for creation to ecology in the vocabulary monastics use today and to what extent we can speak of a charismatic ecology.

*Inquiry*

I have studied contemporary Catholic monastic life in seven countries in Europe, Argentina and five countries in Africa. This article is based on multilocal empiric inquiries with half-structured interviews and participant observations in several Catholic monasteries. Field inquiries were carried out between 2004 and 2019, mainly in communities from the Benedictine and Trappist orders, but also Poor Clare and Carmelites nuns' monasteries. I conducted more than 300 interviews with monks, nuns, abbots and abbesses of these communities. As I seek here to show the evolution concerning the ecological question in monasteries during these last two decades, I will use the interviews from 2004 and compare the different periods between them. I will of course not use the data from all the surveys. The monasteries mentioned are presented in Table 1.

**Table 1.** Main data about studied monasteries quoted in this article.

| Country | Monastery | Order | Sex | Foundation | Inquiry |
|---|---|---|---|---|---|
| Austria | Kremsmünster | Benedictine | F | 777 | 2011, 2012, 2017 |
| | Heiligenkreuz | Cistercian | M | 1133 | 2011, 2012, 2017 |
| | Sankt Paul | Benedictine | M | 1091 | 2015 |
| | Steinerkirchen | Benedictine | F | 1938 | 2012 |
| | Seitenstetten | Benedictine | M | 1112 | 2012 |
| Belgium | Maredsous | Benedictine | M | 1881 | 2008 |
| | Westmalle | Trappist | M | 1804 | 2008 |
| | Clairlande | Benedictine | M | 1970 | 2014 |
| France | Cormontreuil | Poor Clare | F | 1220 | 2019 |
| | Saint-Wandrille | Benedictine | M | 649 | 2004 |
| | Saint-Michel | Benedictine | F | 1898 | 2005 |
| | Tamié | Trappist | M | | 2008 |
| | La Pierre-qui-Vire | Benedictine | M | 1850 | 2005, 2006, 2011 |
| Germany | Plankstetten | Benedictine | M | 1129 | 2007 |
| Italy | Camaldoli | Camaldolesian | M | 1124 | 2007 |
| | Praglia | Benedictine | M | 1123 | 2007 |
| Czesh Republic | Vyšší Brod | Cistercian | M | 1259 1945 | 2017, 2018 |
| Benin | Toffo | Benedictine | F | 1970 | 2019 |
| Kenya | Our Lady of Mount Kenya | Benedictine | M | 1979 | 2014 |
| Senegal | Keur Moussa | Benedictine | M | 1962 | 2016, 2017 |
| Togo | Agbang | Benedictine | M | 1988 | 2013 |

## 2. What Affinity between Monasticism and Ecologism?

Originally, the notion of elective affinity (*Wahlverwandtschaft*) came from the German author Goethe's eponymous novel published in 1809, and has since been used by Max Weber in several of his writings on the social sciences. According to Michael Löwy, "elective affinity is a process through which two cultural forms—religious, intellectual, political or economical—who have certain analogies, intimate kinships or meaning affinities, enter in a relationship of reciprocal attraction and influence, mutual selection, active convergence and mutual reinforcement" (Löwy 2004, p. 103). What is the basis of the relationship between the monastery and its natural environment?

## 2.1. Monasteries between Town and Countryside

At its origin, Christian monasticism in Egypt and Syria developed outside the cities, consciously distanced from the centres of social life, choosing the desert as its preferred land. Nevertheless, not all monastic orders developed the same relationship with nature and the rural world. Some orders chose the city as their foundation, no doubt for security reasons when it came to women's communities. Indeed, exclusively female orders such as the Carmelites nuns, the Visitation nuns or the Poor Clares were founded mainly in the city. Others, on the other hand, such as those of the Benedictine family (living according to the Rule of St. Benedict), have anchored their spirituality in a close relationship with nature as a place of agricultural work and origin of resources for the survival of the community. Indeed, St Benedict recommends in his rule that everything necessary for the community's subsistence should be kept within the monastery enclosure or in its vicinity. For instance: "The monastery should, if possible, be so constructed that within it all necessities, such as water, mill and garden are contained, and the various crafts are practiced. Then there will be no need for monks to roam outside, because this is not at all good for their souls." (RB, 66, 6). The rule of Saint Benedict, written in the 6th century and inspired by previous rules, including that of the Master, was written in a disastrous economic context. The ideal of self-sufficiency—never really achieved—provided the monasteries with security and economic independence.

The evolution of monasticism has in several respects distanced the communities from the natural environment. First of all, because of the attraction of monasteries for the security they offer, the possibility of employment and possibly social support, villages have gradually grown up around monasteries, sometimes becoming towns. François Martin, quoted in the introduction, also says of France that "three-eighths of its towns and villages owe their existence to [the monks]" (Martin 1880, p. 82). The monasteries founded in the 20th century on other continents still reflect this example. Thus, when the monks of Solesmes (France) arrived in Senegal in 1962, they founded the monastery of Keur Moussa in a desert of sea sand. "And the desert will flourish" (Isaiah 35:1) was chosen as the motto for the monastery. At present, the village that has grown up around the abbey has 6000 inhabitants, and if we count the 36 villages that make up the commune, we reach 11,000 inhabitants. Another movement is that of the growth of the towns, which has gradually taken over monasteries originally founded outside of towns. The abbey of Saint-Germain des Prés, for example, was founded in 558 in the middle of the fields. The proximity of Paris was an advantage for the monks' trade. Gradually, the city gained ground and took over the abbey, which is now located in the centre of Paris, albeit not in use. Urban developments have therefore in many cases removed the monasteries from their direct natural environment. In addition, historical circumstances such as the French Revolution led to the removal of the communities' land holdings. However, the link with the natural environment remains an important part of the identity of Benedictine monasteries as we will now see.

## 2.2. The Relationship to Nature as Part of the Monastic Identity

According to Genesis, paradise is a garden; the Garden of Eden was enclosed by high walls. But monasteries seek to be a piece of heaven on earth. For this reason, since the Middle Ages, monasticism has tried to reproduce the original garden in its architecture, especially in Cistercian monasteries. Thus, the monastic garden in the heart of the cloister is a garden of symbols, with Jesus as its centre (the fountain), Mary present in the rosebushes, and "flowers and trees full of fruit [symbolising] the results of good deeds and the virtue that itself facilitated their realization" (Kobielus 1995, p. 223). Some monasteries are still known for their gardens, such as the Benedictine Abbey of Seitenstetten in Austria, which has 110 historical species of roses. Newly built monasteries try to integrate nature as much as possible into the architecture of the monastery, to welcome it into the everyday life of the buildings. For example, the Benedictine monastery of Clairlande in Belgium, founded in 1970 as a reaction against the consumer society, was consciously built in the middle

of the forest with a minimalist architecture that integrates the monastery into the forest. Bernard Sawicki, studying monastic architecture and nature by the monastery of Clairlande writes: "The monastery is built—or rather—hidden in a forest. Entering it is the same as entering the forest; entering the forest one is surprised by entering simultaneously, the monastery." (Sawicki 2019, p. 109). The new Cistercian monastery Val Notre-Dame in Saint-Jean de Matha (formerly located in Oka), Canada, built in 2009, also integrates nature into the monastery with large windows opening onto the forest and light, as well as using wood as the main material. Green roofs complete the integration of the monastery into nature. Under the monastery, a small water treatment plant allows 80% of the waste water to be treated.

Monasticism in the Benedictine tradition has also developed a special role as guardian of the divine creation. The theologian Denis Edwards identifies a specific relationship between the Benedictines and nature, which is a kind of "cultivating and caring for creation" (Edwards 2006, p. 25). "In this tradition, love for God's creation takes the form of responsible farming and preservation of the land. It also involves the love of learning and the conserving of a previous cultural heritage." (id., p. 25). An Italian monk from the Camaldolesian monastery of Camaldoli stated in an interview:

> "Monastic work is born, as a work of contact with the earth, other than as a craft work. The first work is certainly agricultural work and, in any case, some transformation of the goods of the earth. And this is important because the monk has contact with the earth, and therefore with creation, with the cosmos. The role of the human being as custodian, as, how shall I say, promoter of nature." (Br. Amedeo[1], Italy, 2007).

Nature is still the main source of resources for some monasteries. For example, in Austria, the biggest part of the revenue of the male communities comes from forestry. It represents 90% of revenue for the Benedictine Abbey of Kremsmünster, and 80% for the Cistercian Abbey of Heiligenkreuz, for example. The forester of Heiligenkreuz says: "We have two forest companies; one here in Heiligenkreuz with nearly 50,000 ha and other one in Styria with nearly 11,000 ha and half of this area is forest, and the other part is just rocks and mountains." (2017). Nevertheless, the place of natural resources and agriculture in the revenues of contemporary European monasteries is conditioned by the political and religious history of each country which may have led to the suppression and confiscation of property. In some countries, monasteries have been able to keep their land heritage (e.g., Austria), to recover it after a confiscation (e.g., Italy), or, on the contrary, never to recover it in its entirety (e.g., France). For this reason, French monasteries do not have big agricultural activities.

### 2.3. The economy of Stability as Sustainable Development

This linking of Benedictine monasticism to the land is partly explained by the spirituality of stability (*stabilitas loci*), which characterizes monastic orders. Monks and nuns enter a particular monastery and take a vow of stability in that very place. On the contrary, in most apostolic congregations, brothers and sisters enter the congregation and are sent to different stations according to the mission. At the community level, stability means that it must be able to find what it needs to survive in its immediate environment, and this over a period of centuries. We can therefore identify an "economy of stability" (Spalovà and Jonveaux 2018) which is determined by the fact that monasteries are rooted over a long period of time on a secular scale in a specific place. In this sense, it is an economy of sustainable development, in its original sense, needed to meet the requirements of long-term economic development in one place. The objective of sustainable development is indeed "that present generations can meet their needs without compromising the ability of future generations to meet theirs" (Dubois and Mahieu 2002, p. 73). Some monastic economic activities, such as forestry, integrate this long-term vision particularly well. The prior of the Cistercian Abbey Vyšší Brod in the Czech Republic views the economy of the monastery as being a long-term project. He explains that although the forest plots were returned in 2012, it will take time

to earn an income from them, but he says: "We have time: the meaning of the monastic economy is to work for eternity". An employee of the national forestry office who works on a nearby plot of land considers that monks are the best forest owners because they can work on a 100-year scale. The monastic economy is therefore proving to be a sustainable development economy in itself without the need to specifically integrate sustainability as one of its aims.

Stability also implied very early on in monasticism a rationalisation in the use of natural resources which led to innovations both in agricultural techniques and in the production of energy; the water mill, for example. Monastic sustainable development is therefore linked to the rationalisation of the use of natural resources. The aim of this rationalisation is the maximisation of resources combined with a minimisation of their destruction for greater sustainability, as well as a minimisation of the time invested in their use to keep most of it available for prayer. According to Max Weber, rationalisation is a characteristic of monastic life; for him the monk is the "human being who lives rationally, who works methodically and by rational means toward a goal, namely the future life" (Weber 2021, p. 164).

### 3. Monasteries as Pioneers of Ecologism

Contemporary monasteries engage in ecological practices, such as organic farming or renewable energies. Is this the logical continuation of their age-old relationship with the natural environment?

### 3.1. Ecologism as an Economic Opportunity

The relationship with nature in monasticism is based on respect for creation. In the second half of the twentieth century, this respect for Creation was gradually embodied in ecological activities, such as organic farming and renewable energy. In the first decades, it appears that these choices were not only based on ideological convictions, but also on economic criteria. Indeed, the main reason given by monks and nuns during my Master and PhD. research (2004–2009) for switching to renewable energy was economic: to reduce energy costs. For instance, the hydroelectric power station at La Pierre-qui-Vire Abbey was built at the end of the 1960s by a skilled monk who was trained as an engineer, and who had understood the benefits that the river could offer. The founder of the monastery, Father Muard (1809–1854), had already wished to buy both banks of the river to build a mill. Moreover, independent energy production is in line with the idea of economic self-sufficiency as written in Benedict's rule. However, although the motivation for constructing the mill was partly to meet the high electricity needs of the printing house in the monastery, selling a part of the electricity produced was also part of the plan, according to the cellarer. The ecological aspect, as understood today, was not a determining factor when the power station was built. This has, however, become more important over time. Brother Etienne, who is in charge of the plant, states: "I think we have to be concerned about our planet, which is in a bad state, if it goes on like this, it will not last forever. [ . . . ] We have to leave it to the next generations . . . This was not the primary concern of those who built the micro power plant in 1969, it is more so today." (2006). Although this monk is convinced of the ecological value of the hydropower plant, he does not neglect the pragmatic and economic aspects:

> "I would insist much more on the fact that it is an energy that can gain value in the future, because, I always come back to Brussels, there is a directive from the European Commission that will oblige the States to consume 21% of renewable energy. This means that our electricity, produced from water, is likely to have a certain value [ . . . ]." (Br. Etienne, France, 2006).

The ecological arguments are here aligned with the economic arguments; the additional costs of ecological development are therefore not justified. At the Benedictine Abbey of Maredsous, Belgium, the monks have installed solar panels to provide hot water in order

to reduce the electricity bill. In the Trappist Abbey of Tamié, France, monastic ecologism also remains within a rational economic framework:

> "We stopped the wood boiler, for example. So, we still have the wood boiler, but it is a job to go and collect the branches, to saw them, it is better to work at the cheese factory anyway because it [the wood boiler] is not profitable. Today, we have to be realistic, that's all. We must not be romantic either. There is ecologism and everything, you have to take it into account and at the same time be realistic. We have to have something balanced, which corresponds to the community and to the talents of each person." (France, 2008).

The example of the organic farm at La Pierre-qui-Vire is an example of economic opportunism converted into a spiritual witness of respect for creation, but which also allows for innovation. The adoption of organic farming dates back to 1969 and follows a period of close collaboration with the National Institute for Agricultural Research (INRA). The cows on the farm at La Pierre-qui-Vire had all been made ill by various products being tested on them, so it was urgent—especially for the economic situation—to find a solution. In the interviews, the brothers make it clear that the economic disaster caused by the situation was the most important aspect of the problem: "The cows were producing half as much, they were sick, we had incredible veterinary costs. And to such an extent that we said, this is not sustainable, it was not bringing in anything, it was costing the monastery." (Br. Frédéric, bursar, 2006). Another monk says: "All the cows were sick, we had to change the herd, . . . well, we went organic, it was a gamble, and the gamble was won." (Br. Etienne, manager of the hydroelectric plant, 2006). According to today's monks, only two or three of the monks at that time were convinced by the organic solution. The others were only interested in the economic side. The farm thus became the first organic farm in Burgundy, and what was a gamble at the end of the 1960s turned out to be a success, as shown by the current enthusiasm for this label. The term "gamble", used by Brother Etienne, highlights the opportunism of this business. From this economically motivated decision, the monks then constructed an extramundane meaning for it. The commitment to organic farming is therefore seen as a religious act of protecting creation out of respect for the creator.

In these examples, which date from before 2010, ecological sustainability is often a secondary concern for the monastics, with economic reasons taking priority. However, to claim that the ecological commitment of the monastics is only based on economic motives would not be accurate, as ecological aims are also important for monastics. They tend to opt for ecological choices when they can afford to, even though it sometimes means losing money. For example, at Tamié in 2008: "Overall, in the small farm here, we don't use any fertiliser, we try thermal weed killers. Sometimes it is very expensive, it is not possible, but we have that." When asked if they are willing to do this even if it costs them more, the monks answer in the affirmative. However, this small farm has virtually no role in the monastery's economy, suggesting that rationality prevails in the case of activities essential to the community's survival. As we will see in the next section, purely ecological arguments are becoming increasingly more important for the monastics.

### 3.2. Monasteries as Pioneers of Ecologism

As monasteries have always been privileged places of innovation (Jonveaux 2019), and have always integrated sustainability and the conservation of nature into their relationship with the economy and the environment, they are able to establish themselves as pioneers in the field of ecologism in the 20th and 21st century, for instance through organic farming. According to the monastics interviewed, organic farming is a continuation of monastic agriculture. For instance, the French women's abbey of Saint-Michel, after having had the first off-ground turkey farm, was a pioneer in organic farming and owned the first organic farm in its region. The sister bursar said that they had "always been organic . . . before anyone talked about it" (Saint-Michel, 2005). For some monastics, organic farming is the most accomplished form of monastic agriculture. The same sister continues to say: "I am

much more comfortable with the organic look". Many monasteries had a pioneering role in the pattern of ecological production. The director of the brewery in Westmalle, Belgium explains: "They built a water purification plant in the 1960s. At that time, it was really revolutionary." The examples would be too numerous to mention, but the methanisation at Tamié Abbey begun in 2003 is interesting because it is embedded in the latest technologies. Here, again, the original idea was to find a solution to an economic problem. Cheese making produces whey, which is very polluting for the environment:

> "Originally, the whey was given to feed the pigs and then it disappeared, anyway, it was more expensive then . . . Then we had to sell the serum, but we were too far away, so we had to look for a storage area nearer. At first, we were paid 15 cents a litre and at the end when I was in charge of the cheese factory, we had to pay 15 cents a litre to take it away. That's when my successor launched this study on methanisation. And it is a business that works very well because the gas produced provides us with all our domestic hot water. So, there was a lot, a lot of publicity about this business because a lot of people are interested in it [ . . . ] So, there was a lot of publicity, TV, radio . . . " (Bursar, France, 2008).

The main reason for the methanisation is therefore economic, but the monks quickly emphasized the ecological importance of this activity. This gas production allows them to produce hot water every day for 60 to 80 people. This initiative has been widely communicated, including by documentaries filmed by national TV stations. The monastery therefore serves as a showcase for ecological initiatives. Three cheese dairies in the area have already opted for this system thanks to the example of the abbey.

This pioneering role extends to the present day with, for example, the abbey of the German Benedictine missionaries in Munsterschwarzach, which is the first carbon negative monastery in Europe: "The Munsterschwarzach Abbey in Bavaria is not only Europe's first carbon-neutral monastery, it has actually been carbon negative—that is, its activities have a net effect of removing carbon dioxide from the atmosphere rather than adding to it—for nearly 20 years now."[2] This role also extends to Africa, where monasteries are becoming pioneers in the ecological field, whose issues are becoming increasingly important. For example, at Keur Moussa Abbey, the monks are developing organic farming and have developed a vocational training school for agro-ecology. They are therefore also trying to spread ecological awareness to the outside world. In the Benedictine monastery of Toffo, Benin, the sisters consider themselves responsible for the protection of creation. The prioress explains: "And we see to the reforestation. And then to save some rare species. To save some rare species, that is our project. For monastic life in Africa, it is our project to save rare species, pharmaceutical species." (2019). She therefore regards the protection of certain species of plants as being an integral part of monastic life. The role of monasteries as pioneers in the field of ecological innovation is illustrated by a situation experienced by the monastery of Keur Moussa. In 2016, the president of Thailand offered the president of Senegal a solar dryer with innovative technology from the University of Silpakorn. After contemplating what to do with this gift, he decided to give it to the monastery of Keur Moussa, and they have been using it to dry fruit ever since. The exemplary role of the monks is thus recognised within society and its highest authorities. In some cases, difficulties in accessing energy or water are pushing monasteries to find sources of production which are in line with the renewable and sustainable approach. In Kenya, monks developed solar energy and water recycling, meaning that regions without access to mains electricity are able to have a source of electricity. Agbang Monastery also has a solar power source, although in 2013, when I carried out my survey, it did not provide electricity until the evening on cloudy days. The monasteries are recognized in their role as pioneers because their special position in relation to society gives them visibility and credibility, and enables them to also be recognized by the authorities.

*3.3. A Systemic Approach or Rational Ecologism*

The monastic approach is generally a fully coherent systemic approach, or at least it tends to be. "One calls Utopia every total ideological system aiming, implicitly or explicitly, through a call to the imagination alone (written Utopia) or through some transition of practice (practised Utopia), to radically transform the existing global social systems." (Séguy 2014, p. 287). This systemic approach is found in particular in the organisation of monastic places and the resources necessary for subsistence (autarkic approach), but also in the personal asceticism of the monastics which rationalises the whole of daily life (Jonveaux 2018). The systemic approach to ecologism is actually embedded in the general rational approach to monastic life and its relationship to the world.

In practical terms, this means that ecological monastic practices do not stand alone, but rather are part of a set of mutually reinforcing practices aimed at the same goal. For example, since 2006, La Pierre-qui-Vire Abbey has had a wood-fired boiler in operation as a result of renewed ecological considerations. The ecological and economic reasons for this are of equal importance and form part of the local system, which the monks are very keen on. A brother explains that by so doing, they are participating in the local economy, as they are using the local wood industry and sawmills. In addition to this, the monks are thinking of using the ash as fertiliser for the organic farm. This is a well-thought-out project involving a consultancy firm to evaluate its profitability. The same integral and systemic approach can be noted in an interview with a forest brother from Heiligenkreuz:

> "I think, we do not start with the wood. We start with planting trees, or with natural regeneration of the trees. And from this point to the end, when we harvest the wood, I think the whole process should be good. [ . . . ] When you use sustainability, you can speak about three sectors—the economic, the ecologic and the social part. And I think, in these three parts we should provide a good example in our country, to other firms and companies." (Forester, Austria, 2017).

The monastic system aims to propose a coherent approach between the economic, the social and the ecological, which corresponds to integral ecology. This approach fulfils Pope Francis' encyclical *Laudato Sii* (2015), as integral ecology takes the environment into account as much as the social aspects.

An example of a coherent ecological system in a monastery environment is the monastery in Plankstetten, Germany, which, when studied in 2008, had already developed an "*Öko-autarkiekonzpet*" (ecological autarky concept), which they continued to develop. Every single dimension of the monastery is integrated into the "*regionales Autarkiekonzept*": organic farming, organic gardening, renewable energy, etc. This *Öko-Koncept* (ecological concept) is constructed in such a way that all of the monastery's actions are carried out with an ecological focus. This integrates all the dimensions of monastic life: food, production, energy, fuel for the cars, reception of visitors, etc. The graphic representation of this organisation proves its systemic dimension. At the centre of the circle is the kitchen, which receives the monastery's products (bakery, butcher's, garden, farm, agriculture), transforms them and redistributes them into meals for the community, the hotel and the brewery. The second concentric circle, around that of the monastery, includes other organic producers in the region, who are especially necessary for the beer. The monastery is heated mainly by biomass, with wood chips coming mainly from the monastery's carpentry. Water is presently heated by 60 m$^2$ of solar panels, and other small buildings are equipped with decentralised natural gas burners to avoid heat conduction losses[3]. Electricity is produced by photovoltaics and biogas from fermented livestock waste. In this way, the monastery is self-sufficient in energy. The monks emphasise that they seek to use raw materials with as little waste as possible, reusing the residues for other activities, which for them corresponds to a responsible relationship with creation.

## 4. From Respect of Creation to Ecologism

The eco-friendly practices of the monasteries are therefore not necessarily new in their application, but are now integrated into another symbolic framework originating in society, without denying the old framework of respect for divine creation. Monasteries are currently engaging in practices that they openly describe as ecological and sustainable and are thus gaining a new plausibility in societies where Christian culture is losing momentum.

### 4.1. A New Vocabulary for the Same Activities?

Between the beginning of my investigations in 2004 in France and today, the ecological discourse of monks and nuns has evolved considerably. I had already identified (Jonveaux 2011a) disparities between France and Germany, the latter having been interested in ecological issues for a longer time, especially on the political level. At the time, this discourse already had a place within the German Catholic Church, and the Abbot of Plankstetten, who had become a bishop, was described in the media in 2008 as a "green bishop". Ludovic Bertina identifies a "delay" in French Catholicism concerning ecological issues. "Whereas in Germany, the bishops as a whole published Being Responsible for Creation as early as 1985, and in the United States, Renewing the Earth was published by the Bishops' Conference in 1991, it was not until the year 2000 that several French bishops jointly supported a fair understanding of the ecological struggle." (Bertina 2017, p. 35). In Latin countries, ecological commitment and religious commitment are two separate "arenas" (Neveu 2015) in the public space, to use a term from the sociology of social movements. The prophetic proclamation of ecology by monastics was unusual, as environmental activism and religious prophetism were mutually exclusive; indeed, environmental activists were often anticlerical. In the years during which the Church did not address this subject, the monastics were responsible for bringing a religious perspective to the ecological arguments, by acquiring a place in the public arena as religious ecological activists. One could already identify expectations of society towards the monks concerning environmental issues. Thus, in 2004, the bursar of Saint-Wandrille Abbey in France, referring to his wish to make the monastery energy independent, said: "It's a matter of choosing either a nuclear battery or a hydrogen battery … or a wind turbine, but the wind turbine (laughs), nuclear is the easiest, it already exists. But the problem is more of a communication problem. The Abbey of Saint-Wandrille uses nuclear energy! (laughs)" (Br. Denis, Bursar, 11.04). The communication of monastic activities is therefore at stake and must be integrated into a system of plausibility expected by society.

This situation has changed radically since then, particularly with the publication in 2015 of Pope Francis' encyclical *Laudato Sii*. The appropriation of ecological themes and their semantics by the Catholic Church has led to an evolution in monastic discourse which has integrated secular ecological semantics and arguments. This allows monastics to position themselves in this field and to make them plausible in societies that are increasingly less interested in eschatological salvation. The feared end of the world at the beginning of the 21st century is essentially an environmental apocalypse for which man, not God, bears responsibility.

The monasteries are currently progressing from opportunistic ecologism, which involves economic arguments and responding to current demand from society, to an ecologism of commitment. For example, an Italian monk from Praglia acknowledges: "It is a bit of a fashion to work with nature" (2007), which lends credence to the fact that ecological commitment is not determined solely by religion. However, the commitment of the monastics to the environment can only be seen as opportunistic insofar as its inclusion in the ecological arena is concerned, since the protection of creation is intrinsic to monastic values. The opportunistic approach is also a response to a demand from society that corresponds, in a mirror image, to the offers made by the monastery. A monk from Camaldoli explains: "In today's context, the environment is becoming a requirement for every man and woman, and this is also being recovered at the monastic level, contact with nature, in

the laborious sense of the word, not just contemplative, is becoming a way of keeping up with the stronger demands of today's man." (Br. Davide, Italy, 2007).

This change in approach to environmental concerns and ecology can be recognised by the lexicon used. The concrete actions of monastic communities may remain the same, but the opportunistic arguments, especially economic ones, are now less common. However, despite an asserted ecological commitment, some monks "do not like to use this term 'sustainable' because they have the impression of being dispossessed of something which was traditionally monastic before it became a societal concern." (Spalovà and Jonveaux 2018). A Cistercian monk from Heiligenkreuz says: "When you talk about sustainability—I don't use the word, I don't like to use it, because I think it is a very 'nobody knows what it is'. So, I think it is better to use other words, more detailed." (2017). But in general, the environmental actions of monasteries are currently verbalised as such and in terms that come from the ecological scene. A former abbot from La Pierre-qui-Vire said in an interview in 2011: "The famous ecologism nowadays . . . before we didn't talk about it, but we lived it too." The relationship between the monastics and nature has not changed a great deal; what is different is the way in which this relationship is verbalized and presented to society. A shift can also be observed in the association of monastic identity with ecological commitment. In 2006, a monk from La Pierre-qui-Vire said about renewable energy: "We don't do this as part of our monastic life. We are sensitive to this dimension of safeguarding nature and preserving natural resources and it is true that we try not to do anything by wasting energy." This monk did not associate his ecological action with his identity as a monk. In 2015, a Benedictine monk from Sankt-Paul in Austria, who is careful to develop his honey business organically, said the contrary: "This responsibility for creation, that is our mission as Benedictines." Or as a Benedictine sister in Austria explains: "We have a woodcut heating here in the monastery, that is together with the parish. There is also the issue of creation responsibility in the church. Getting people to use green electricity. And after that, energy must also be saved." (Steinerkirchen, 2012). As environmental and climate issues become more and more prominent within society, the monastics identify acting upon this as being one of their *responsibilities.*

*4.2. New Opportunities for Monastic Products*

Society's interest in ecological issues is an opportunity for monasteries, while their purely religious dimensions are attracting fewer and fewer people. The natural quality recognised in monastic products resonates with the current demands of society. Monastic products have therefore become increasingly popular in recent years, particularly in Europe. A monk from La Pierre-qui-Vire said in 2005: "We need to sell products that show respect for creation and for people at the right price." And the manager of the shop of the Abbey of Maredsous observes: "In other words, monastic consumer products have a good image because they are associated with an image of quality, natural products, etc. and so we have built up a loyal clientele around these products." (2008). As mentioned, not all monastic products carry the organic logo. This label implies costs that the communities do not necessarily want to pay. However, the production methods are nevertheless environmentally friendly even if they are not certified with the organic logo.

The public already converted to monastic products is not necessarily looking for an organic logo as well. Trust in natural monastic production methods is sufficient when it is established that the product is truly monastic. The public less familiar with monastic reality is more attached to organic labels. For example, the herbal teas and mixtures produced by the Gut Aich monastery in Austria for the Spar supermarkets, and which are therefore aimed at a broad public, carry the organic label. The product range is called "Wie Früher" (like before), which gives confidence, although the monastery itself was founded in 2004 and is thus the youngest in Austria. On the French website for online sales of monastic products "Artisanat monastique", a specific tab is dedicated to organic products[4]. It lists 105 products, including 57 food products and 48 cosmetic or wellness products (essential oils for example). Of the 505 food products offered for sale, 263 carry only the Monastic

label, which certifies products made by monastics in a monastery, 21 only the organic label and 46 both labels. This shift towards ecology can also be seen in the criteria of the French brand Monastic, which since 2018/2019 has included ecological criteria summarised as follows: "A commitment to sustainable development and integral ecology", with a reference to *Laudato Sii*. Ecology is expressed and affirmed as such in monastic products, following on from the expectations of consumers. The ecological commitment of monastics to sustainable development therefore responds on the one hand to the Catholic values of respect for creation and on the other hand to the expectations of today's society for whom ecological and climate disaster are of great concern.

*4.3. A Charismatic Ecologism?*

In the field of ecologism, monasteries are currently in competition with secular institutions. What is the specificity of monastic ecology in society? Where does the particular confidence in monastic ecological activities or products come from?

First, as previously mentioned, monasteries have a centuries-old tradition of relationship with nature based on respect, sustainability and rational use of resources. However, monastics are particularly well known for their ecological commitment, not only because they have been experts in this field for centuries, but also because their inimitable identity gives them a different relationship to these realities. In other words, the reputation of ecological monastic practices comes from the charisma associated with the monastics. According to Max Weber, charisma is the "extraordinary quality [ . . . ] of an individual who is, so to speak, endowed with strength or characteristics which are supernatural, superhuman, or at the very least outside of everyday life, inaccessible to the average person, or alternatively who is considered to be sent by God or as an example, and consequently a 'chief '" (Weber 1995, p. 320). For charisma can only exist if it is recognized within society: "Sociologically speaking, holders of charisma can only be so described if their 'gift' or 'gifts' are recognized as such by a group of disciples or followers" (Séguy 2006, p. 141). In secularized society, purely religious charisma seems to have been devalued, but we can observe that monks and nuns are recognized in their specific commitment for ecology.

Before ecology became a central issue in society, monasteries played a prophetic role in this field. Prophetic means that they tried to announce it when these subjects were not yet very widespread in society. Monastics do not aim to keep their ecological commitment behind the walls of their monastery, but to be witnesses to society of their environmentally friendly practices. A brother from Plankstetten in Germany said in 2007 that they have been "prophesying for the organic-boom (*Bio-Boom*) for some years". The monks opened an Ecological Information Centre for the public with an exhibition and a range of events. This aim of educating people's consciences on ecological issues—and not only or not directly religiously—is taken up by a forest brother from Heiligenkreuz: "Also with the visitors of the forest we want to show the possibilities of how to use the forest in a good way." Since the publication of the encyclical *Laudato Sii*, ecology has also become the subject of spiritual retreats, as offered, for example, by the Poor Clare Sisters of Cormontreuil in France in 2022.

Because of their particular identity due to their lifelong commitment, monastics carry a trust also associated with their ecological practices. The monk responsible for beekeeping at the monastery of Sankt-Paul in Austria said:

> "Meanwhile, I still believe that for people it is somehow something special, something different. A normal person can produce good or better honey, but it is general with monastery products, also monastery wine, it somehow still conveys something like exclusivity. [ . . . ] There is also more trust somehow." (Sankt-Paul, 2015).

We are currently living in a context of mistrust of food products. "Organic is [then] conceived as a possible (though not fully satisfactory) answer in a context of health uncertainty: the organic product is potentially a safe product" (Lamine 2008, p. 121). However, if this organic product is recognised as safe, there may be doubt about the authenticity of

the label because "the possibility of merchandising differences thus opens a new era of suspicion" (Boltanski and Chiapello 1999, p. 539). Even organically grown products are not necessarily trusted, even labels are subject to distrust. In this context, the monastics inspire a particular trust in the authenticity of their ecological commitment.

We can then speak here of a charismatic ecologism, taking up the same characteristics that I had noted for the charismatic economy of monasteries: "Monks' economic charisma thus creates two elements: the extraordinary identity of producers and a reputation for quality, linked to the particular production conditions in unworldly locations and thanks to centuries-old recipes." (Jonveaux 2011b, p. XIII). This charismatic ecologism is based first and foremost on the particular identity of its actors, rather than on labels that may only validate an authenticity that is already present. Because of the trust it inspires, which is based on the identity of the monastics, monastic ecology has a special place in society, especially as a model.

### 5. Conclusions

Benedictine monasticism undoubtedly has an elective affinity with nature, which since the 20th century has become an elective affinity with ecologism. The ecological practices of monasteries, in particular for renewable energies or organic agriculture, are not particularly new. Even in the Middle Ages, hydraulic energy was used. However, these practices are currently reviewed in the light of current environmental issues and monks and nuns appear in this field as pioneers and experts. Monastic ecologism is characterized by two dimensions: the integral and systemic dimension, and the pragmatism of decision making. The monastic ecology is not romantic or ideological, but rational. The main characteristic of the ecology of monasteries is that it is a charismatic ecology, whose reputation and confidence in its authenticity rests directly on the particular identity of its actors, the monastics, elected by God.

Through spreading the ecological message in this way, the monastics are actively participating in changing society for the better, whilst at the same time becoming more plausible in a society which expects ecological solutions but is critical of religious institutions.

**Funding:** This research was partly funded by FWF Lise-Meitner grant number M1271-G15 and by Aktion Österreich-Tschechien OeAD grant number 76p9.

**Institutional Review Board Statement:** Not applicable.

**Informed Consent Statement:** Informed consent was obtained from all subjects involved in the study.

**Data Availability Statement:** Not applicable.

**Conflicts of Interest:** The authors declare no conflict of interest.

### Notes

[1]  The names have been changed to protect anonymity.

[2]  Available online: https://www.americamagazine.org/politics-society/2019/06/18/1200-year-old-benedictine-monastery-has-been-carbon-negative-20-years (accessed on 3 February 2023).

[3]  Available online: https://www.kloster-plankstetten.de/oekologie/energie/ (accessed on 3 February 2023).

[4]  Available online: https://www.artisanatmonastique.com/169-le-bio (accessed on 10 February 2023).

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
