# Peer review of "Monasticism and Ecologism: Between Economic Opportunity and Religious Convictions?"

_religions, doi:10.3390/rel14050575_

Round 1
Reviewer 1 Report
Dear author, Thank you for your efforts. Monastic ecology is an important topic. This has nothing to do with you as a person, but this essay should be completely scrapped and rebuilt. There is nothing about this essay that I would feel comfortable approving for publication. The below comments are just the most important deficiencies which are theoretical, methodological, and organizational. I do hope you resubmit an essay on monasticism and ecology. But not this one.
20 What is the time period for this claim? I don’t think it is at all true that the monks themselves were doing much of that cultivation. They had serfs, renters and lay bothers (Conversi).
23 Ecologism is undefined as well as eschatology. What is elective affinity?
38 Charismatic ecology is what then? More theory than just Max Weber.
The table of data should not be included unless it relates to your specific claims about a region. Its lovely that you have interviewed so many monks and nuns over the years, AMAZING! But tables should display data related to the research project. This was also not a methods section. You needed to discuss how you organized or analyzed the data from your semi-structured (not half-structured) interviews.
51 This section about affinity needs a better subheading and a stronger theoretical background. And the term needs a basic definition earlier in the intro.
114 the example of the monastery in “Oka, Canada” does not even name the monastery. And they actually moved from there, and the modern green structures are part of the new monastery. This example is poorly chosen, poorly placed in the essay, and not accurately detailed.
SEE: https://www.catholicregister.org/item/25146-quebec-trappists-renew-spiritual-life-in-their-eco-friendly-abbey
124 you cant just say “Camaldoli” you have to say what it is and where it is and what kind of monastery.
131 if you have changed the names, why use any at all if its just in parentheses at the end of the quotation? Also, as I tell my students, you can’t just leave a quote hanging, you need to unpack it afterwards.
134 saying that a monastery gets revenue from forestry says nothing about the monks relationship to the forest. They probably contract out the forestry and it is probably for revenue generation. Is there a stewardship plan? Restoration projects?
168 I am not sure if we understand the same thing by rationalization and its not a great connection to nature and ecology. It’s the way we domesticated and dominated the land for money. And even if the monks have a 100 year cycle, they are doing so with this in mind. So if you want to keep this section connect rationalization to some aspect of sustainability, and maybe find a better word for it? Or lose it all together.
188 what is La Pierre-qui-Vire????
240 You do not present a very compelling case that they converted to organic to respect the Creator…
255 how are you showing that monasteries are innovators? Because they developed some medieval farming technology? They are for the most part, conservative, slow to adopt new ideas and very resistant to change.
258 you just have not established anything like the evidence needed to say that contemporary monasteries are pioneers in ecology with one or two who farm organically or installed a water purifier in the 1960s.
The monastery that claims to be carbon negative! Now that’s a cool idea. You should do an entire section on that.
Really the burden of evidence is to show that there is any kind of general trend here. Your surveys should be sent out to as many monasteries as you can find in a particular region, and then see what the percentage of projects, uptake and practices are. All this evidence is selective and cherry picked. And there isn’t even that much of that. The sections need to be organized by activity or impact: Forestry, energy, water, carbon. And then build actual evidence to support more modest claims about monasteries adopting these practices for both practical and spiritual reasons. But the “elective affinity” is ALWAYS in line with the practice needs and theologies of the monasteries.
294 WHERE IS THE MONASTERY?
All footnotes should be formatted according to the style guide and not be floating links.
317 the monks are global utopians? Unconvincing really.
346 now this would make a great case study. This should be its own section where the author highlights the insights of a particular Catholic ecological principle and then uses this monastery as an example of integral ecology. But the data is coming from 2008. Can the author verify what has changed? Or improved?
Unless you are going to build a general theory of monastic ecology using your data, you should narrow in on a few cases that demonstrate a central idea like charismatic ecology, or examples of how Laudato Si has impacted a particular monastery's evolution toward ecology. The sweeping claims about being pioneers, or always attached to nature and the fluttering from case to case does not serve the writer or the reader to make a compelling case for anything like a monastic ecology. Again, you have a great topic but essay does not live up to the topic.
Author Response
Dear Reviewer,
thank you for reading my article. I will take from your comments, what I think good for my article. I agreee with some points, but I disagree with others. Maybe, you could do it with more respect. The table of presentation of the monasteries is intended to give the main information without having to repeat it for each quotation.
Best wishes.
Reviewer 2 Report
Excellent and exciting piece! I found only a few issues of English usage that could be adjusted :
Line/original/suggested revision
19/clear/cleared
67/leaving/living
91/disues/"now abandoned" or "not in use"
98/Garden of Eden/Garden of Eden was
143/That is for this reason for instance/For this reason
149/most of/most
182/therefore/[eliminate]
185/are/were
192/upon/[eliminate]
301/plant/plants
345/the/[eliminate]
344/as integral ecology/eliminate the ","
That's all! Otherwise an amazing article.
Author Response
Dear Reviewer,
thank you for your very positive review! I will check the language and integrate your corrections.
Best wishes
Reviewer 3 Report
a plan of the sites (at least on a European scale) might help the reader
the reference to the rule of St Benedict (lines 70-75) perhaps needs a brief background on the complex and layered genesis of this rule, in order to better define its value
the statement in lines 313-315 needs better argumentation
Author Response
Dear Reviewer,
thank you for your comments. I will improve lines 70-75 (it is a good point) and 313-315.
Do you mean a map of Europe with the different monasteries I am talking about? I will try if I can do that.
Best wishes